# Nanocurcumin Release from Self-Cured Acrylic Resins; Effects on Antimicrobial Action and Flexural Strength

**DOI:** 10.3390/bioengineering10050559

**Published:** 2023-05-08

**Authors:** Parsa Soleymanijadidi, Meysam Moradi, Fahimeh Hamedirad, Zahra Ghanavati, Solmaz Maleki Dizaj, Sara Salatin

**Affiliations:** 1Dental Materials Research Center, Health Research Institute, Babol University of Medical Sciences, Babol 47176-4774, Iran; 2Dental and Periodontal Research Center, Tabriz University of Medical Sciences, Tabriz 51548-53431, Iran; 3Department of Dental Biomaterials, Faculty of Dentistry, Tabriz University of Medical Sciences, Tabriz 51548-53431, Iran; 4Neurosciences Research Center, Tabriz University of Medical Sciences, Tabriz 51548-53431, Iran

**Keywords:** curcumin, nanoparticles, flexural strength, self-cured acrylic resins

## Abstract

The placement of orthodontic appliances into the oral area can lead to infection, inflammatory and gingival collapse. Using an antimicrobial and anti-inflammatory material in the matrix of orthodontic appliance may help to reduce these issues. This study aimed to assess the release pattern, the antimicrobial action and the flexural strength of self-cured acrylic resins after adding different weight percentages of curcumin nanoparticles (nanocurcumin). In this in-vitro study, 60 acrylic resin samples were divided into five groups (n = 12) based on the weight percentage of curcumin nanoparticles added to the acrylic powder (0 for control, 0.5, 1, 2.5, and 5%). Then, the dissolution apparatus was used for the release assessment of nanocurcumin form the resins. For antimicrobial action assessment, the disk diffusion method was used and a three-point bending test was performed with a speed of 5 mm/min to determine the flexural strength. Data were analyzed using one-way analysis of variance (ANOVA) and Post-Hoc Tukey tests (with *p* < 0.05 as significant level). The microscopic images showed the homogeny distribution of nanocuricumin in self-cured acrylic resins in varied concentrations. The release pattern showed a two-step release pattern for all concentrations of nanocurcumin. The one-way ANOVA outcomes indicated that adding curcumin nanoparticles to self-cured resin increased the diameter of the inhibition zones for the groups against *Streptococcus mutans* (*S. mutans*) significantly (*p* < 0.0001). Additionally, as the weight percentage of curcumin nanoparticles increased, the flexural strength decreased (*p* < 0.0001). However, all strength values were higher than the standard value (50 MPa). No significant difference was detected between the control group and the group with 0.5 percent (*p* = 0.57). Considering the proper release pattern and the potent antimicrobial activity of curcumin nanoparticles, then the preparing self-cured resins containing curcumin nanoparticles can be beneficial for antimicrobial aims without damaging the flexural strength to use in orthodontic removable applications.

## 1. Introduction

With a growing number of people looking for orthodontic material usage, it is now common to cure patients who require adjustment of periodontal disease. The relationship of periodontic–orthodontic issues has been an interesting topic for a lot of studies until now, and is a still contentious subject [1]. The addition of orthodontic appliances into the oral area may cause infection, inflammation and gingival failure. Even though orthodontic treatments recover dental and skeletal issues, the location of an orthodontic appliance in a patient’s oral cavity may also lead to changes in their oral hygiene practices and affect their periodontal health [1,2].

Self-curing acrylic resins are used for producing removable orthodontic appliances [3]. Some reports on orthodontic appliances show an increase in *Streptococcus mutans* (*S. mutans*) [4,5]. Moreover, removable orthodontic appliances may expand dental caries. So, the application of antimicrobial materials such as chlorhexidine mouthwash is recommended in orthodontic patients since mechanical clearing is not able to eliminate the microorganisms from all plaque-retaining parts [6]. Studies also have revealed main tooth decay and gingival inflammation problems after 4 months of eating and drinking without cleaning in patients using an orthodontic appliance [5,7,8]. Therefore, the cleaning of orthodontic appliances is vital to maintain oral health. Then, using a multifunctional (antimicrobial, antifungal and anti-inflammatory) material in the matrix of orthodontic appliance may help to reduce these issues.

To one side, the antibacterial activity and, to the other side, the mechanical possessions of acrylic resins are similarly important [8]. Flexural strength is one of the main properties of resins defined as the maximum force a material tolerates before being bent [9]. The minimum limit of flexural strength for dental polymers has been defined in ISO 20795-1 (2008) that should not be <50 MPa [10]. Thus, investigators stated that the impacts of additives on the flexural strength of acrylic resins should be evaluated to avoid the reduction of their strength to lower than the standard value [9].

Curcumin is the active ingredient of the *curcumin longa* plant. It was used traditionally for color, flavor, and medicinal purposes. Curcumin has different therapeutic applications in treatment of diabetes, pancreatitis, arthritis, neurological disorders, and different cancers [11,12]. Since this material has certain different properties (i.e., antioxidant, antimicrobial, and anti-inflammatory), it would be beneficial in dentistry. Curcumin has been used in fissure sealants, mouthwash, dental plaques diagnosis, and so forth [13].

A nanomaterial refers to a material that has at least one dimension in a three-dimensional space or is reduced in composition to a nanoscale (1–100 nm) [14,15,16]. Nano-dentistry is the usage of nanomaterials in the dentistry field [17]. Nanomaterials have broad and promising applications in future dentistry. When selecting a nanoparticle for use in the arena of nano-dentistry, its chemical, physical and biological features of its nanostructures are taken into account. The clinical usage of curcumin is restricted due to its low bioavailability. Moreover, data around its safety in higher doses are rare. Curcumin nanoformulations are being revealed to improve curcumin’s bioavailability and decrease its toxicity [17,18]. Indeed, nanoparticles have unique belongings that are not found in conventional macro- and microscopic ones.

Curcumin can prevent bacteria function by destroying the bacterial membrane [15]. Since nanoparticles are more effective in diffusion across the cell membrane of bacteria due to their smaller size, they can be used against pathogenic bacteria [19,20]. There are some studies about curcumin’s properties (i.e., its anti-corrosion property, adhesion improvement, and flexibility enhancement) [21,22]. Furthermore, curcumin’s effect on the shear bond strength of orthodontics composites has been studied [23]. Adding one percent weight of curcumin nanoparticles to a composite showed a significant antimicrobial effect on cariogenic bacteria without affecting the orthodontic composite shear bond strength of bovine teeth [23]. Then, regarding the proven antioxidant, antimicrobial, and anti-inflammatory properties of curcumin, the current study aimed to investigate the effect of the curcumin nanoparticle on the flexural strength of self-cured acrylic resins.

## 2. Materials and Methods

### 2.1. Materials

Self-cured acrylic resin was prepared from Acropars Medical Co. (Tehran, Iran), curcumin nanoparticles was purchased from Golestan Co., Tehran, Iran. Carbide silicon papers (grit 240) was bought from Xinya Co., Changzhou, Jiangsu, China. Phosphate buffer was prepared from Sigma Aldrich Co., Saint Louis, MO, USA. *S. mutans* was purchased from the Pasteur Institute of Iran (Tehran, Iran). Müller Hinton agar was prepared form Sigma Aldrich Co., Saint Louis, MO, USA. Dental pulp stem cells were bought from Shahid Beheshti University (Tehran, Iran).

### 2.2. Methods

#### 2.2.1. Sample Preparations

The self-cured acrylic samples were separated into five groups (n = 12) based on the curcumin nanoparticles’ weight percentages (0, 0.5, 1, 2.5, and 5). The curcumin nanoparticles weight were measured using a digital scale (Shimadzu LIBROR AEU-210/Germany Sartorius) with an accuracy of 0.0001 and mixed with acrylic powder in the weight percentages of 0.5, 1, 2.5, and 5. To obtain an homogeneous mixture, after manual mixing, it was mixed in an amalgamator device (Dentine 4, Farazmehr, Isfahan, Iran) for 20 min. Afterward, the monomer was added to each group of sample powders (3.5 g monomer + 5 g powder) for 30 s, according to the manufacturer’s instructions. Approximately one minute after mixing, the resultant paste was packed in the Doghy stage in the prepared metal molds that was in the standard dimensions of the flexural strength test specimens [24]. To prevent monomer evaporation, bubbling, and the mixture becoming clear, the metal mold was put in a pressure cooker with a pressure of 20 psi [25], and then the samples were taken out and polished. The samples were polished by carbide silicon papers [24]. Table 1 shows the formulations for the samples.

#### 2.2.2. The Distribution Pattern of the Nanoparticles in the Acrylic Samples

The size distribution for curcumin nanoparticle was measured by Particle Size Analyzer (PSA, Horiba Scientific SZ-100, Shanghai, China).

Transmission electron microscopy (TEM) is a potent instrument to examine the distribution pattern, the interaction, the structure, and the morphology of nanoparticle-based systems. To investigate the distribution pattern of the nanoparticles in the acrylic samples, a sample of each group was examined under an electron microscope (TEM, NeoScope, Nikon Instruments, Melville, New York, NY, USA). A trained laboratory technician prepared the samples. Samples were stained with 2% tungistic acid for 15 min, stopped on copper grids, and dried overnight for observing by TEM. For TEM analysis, 3 samples were chosen randomly from each group. Then, the best-quality figure was chosen.

#### 2.2.3. Release Study

Drug dissolution device 2 was applied to measure the release outline of nanocurcumin release from the resins. This device is in fact the most extensively used technique in drug release examining, particularly for nanomaterials. It contains 6 holes (wells). For this test, 300 mL phosphate buffer was transferred into each well of the apparatus (temperature of 37 °C, pH of 7.4, and a stirring rate of 100 rpm). Then, 5 mg of the samples was used. One-milliliter liquid samples were taken from the wells every day and the absorbance was observed by means of a UV spectrophotometer for curcumin at 350 nm [17].

For method validation, stock solutions of curcumin were prepared in methanol (standard dilutions varying in 1–10 μg/mL). Least square regression analysis was performed for the gained results. The validation results showed good linear correlations between the absorbance and the concentrations of 1–10 μg/mL (regression coefficient of 0.9991, correlation coefficient of 0.9995, limit of detection of 1.05 ng/μL, and limit of quantification of 3.11 ng/μL).

#### 2.2.4. Antimicrobial Test

The unique technique of defining susceptibility to antimicrobials is based on broth dilution methods. In this study, the disk diffusion method as a routine laboratory examination was used to examine the antibacterial effect of curcumin containing self-cured acrylic samples. This technique recognizes the action of bacteria on an antimicrobial material by making a gradient of concentration around a disk.

*S. mutans* was purchased from the Pasteur Institute of Iran (Tehran, Iran). Then, the disk diffusion method was used to examine the antimicrobial action for the studied groups (0 for control, 0.5, 1, 2.5, and 5%). Mueller–Hinton agar was applied as the culture media. Sterilized swabs were flooded in microbial solution at a concentration of half McFarland (1.5 × 108), and then lawn cultivation was led three times on the plate. After 24 h of incubation at 37 °C, the prepared samples (as disks) were placed on the culture media. Then, after 24 h of incubation, the plates were assessed for the inhibition zones neighboring the disks [17,18].

#### 2.2.5. Determining the Cytotoxicity

Cell viability determination using 3-(4, 5-dimethylthiazolyl-2)-2, 5-diphenyltetrazolium bromide (MTT) was carried out to obtain the cytotoxicity of samples against dental pulp stem cells. Samples in disk shapes with diameter of 5 mm was located in the end of the cell wells and then the cells were cultured in a single layer in Dulbecco’s Modified Eagle Medium (DMEM) containing serum and antibiotics.

After 72 h, the cells were washed and incubated with 200 microliters of culture medium along with 50 microliters of MTT solution (2 mg/mL PBS) for 4 h at 37 °C and away from light. Then, the overhead solution was removed and 200 microliters of DMSO and 25 microliters of Sorenson glycine buffer was added to each well. Then, the absorbance was read at 540 nm and the percentage of living cells was assessed according to ISO 10993-5 [26] by comparing the control (cells grown without any material).

#### 2.2.6. The Flexural Strength Test

Flexural testing is utilized to define the flex or bending properties of a material and includes insertion a sample between two points or supports and starting a load using a third point or with two points which are, respectively, called 3-point bend and 4-point bend testing. Maximum stress and strain are determined on the incremental load applied. Outcomes are exposed in a graphical format with tabular grades counting the flexural strength (for fractured samples) and the yield strength (samples that did not fracture). The usual materials tried are polymers, plastics, composites, metals, ceramics, and wood.

In this experiment the number of samples for the flexural strength test was determined to be 60 based on a similar study [24] and the statistical analysis. Samples were fabricated according to the ISO 1567 standard in the form of a rectangle with dimensions of 60 mm × 12 mm × 4 mm [24]. Figure 1 shows the samples.

To perform the flexural strength test, the universal test device (Koopa, Iran, Sari) was used. A blinded, trained operator performed the bend test steps; 3-point flexural strength was performed with a speed of 5 mm/min. The applied force on the samples increased until they fractured. The maximum force that appeared at the time of fracture was recorded (Figure 2). The flexural strength was calculated using the following equation.
$$ơ=\frac{3WL}{2bh^{2}}$$

The ơ is flexural strength in MPa, W is the applied flexural force in *N*, L is the distance between two different positions of the samples seated on the instrument in mm, b is the sample width in mm and *h* is the sample thickness in mm [24].

#### 2.2.7. Statistical Analysis

The Shapiro–Wilk test showed the status for the normality of data. Then, we used SPSS 20 (IBM Company, Armonk, NY, USA) to compare the data between groups with a *p* value of <0.05 as a significant level through one-way analysis of variance (ANOVA) and Tukey–HSD post-hoc tests.

## 3. Results and Discussion

The signs realized clinically after the placement of orthodontic appliances into the oral area can lead to certain primary problems. Boke et al. stated that in patients cured with fixed orthodontic appliances, observable plaque, noticeable inflammation, and gingival collapse presented important upsurges after orthodontic treatment. In their study, the relationship between the lower incisor position and gingival collapse in patients cured with a fixed appliance and removal was positive. Furthermore, throughout the orthodontic treatment, the mean amount of observable plaque and visible inflammation presented important upsurges. Thus, earlier than orthodontic treatment, in the case of patients with a high degree of periodontal health, it is necessary preserve it during the treatment phase. Therefore, lower incisor inclination alteration should be assessed with more measured prospective training throughout the orthodontic cure to stop damaging side-effects [2].

### 3.1. The Distribution Pattern of the Nanoparticles in the Acrylic Samples

The low bioavailability of curcumin is the most significant worry for its clinical usage. Additionally, little data is obtainable on its safety at higher doses. Recently, to reduce its toxicity and improve the bioavailability of curcumin, new strategies based on nanomedicine have been exposed. The small size of the nanoparticles is also beneficial as it enables their diffusion into ultra-small hovels, indentation, and capillary areas in the polymer matrix. Dropping the size of nanoparticles increases its surface area, the interaction of these nanoparticles with the environment upsurges, and the means of crossing the body barriers and going into its cells them will be different.

The size distribution for curcumin nanoparticle is shown in Figure 3A. The particles showed a mean size of 85 nm. The TEM images show the distribution of nanocuricumin in self-cured acrylic resins in varied concentrations (Figure 3B). The image showed that nanoparticles had good distribution on a self-cured acrylic matrix. As the concentration of nanoparticles increased, the aggregation points of nanoparticles on the resin matrix increased. The content of 2 wt% of nanoparticles were better spread in the matrix. Similar findings are also reported by other researchers [27,28,29].

### 3.2. Release Study

Figure 4 shows the release outline of the nanocurcumin from resins. The samples presented a two-stage release outline: a rather fast release form in the first 10 days for curcumin and a constant sustained release until the 30th day. Based on the reports, the quick release of curcumin is related to the curcumin particles in the outer surface of the resins and the continuous release is associated with the nanoparticles that are more internal and may have electrostatic connections with the resin matrix [17].

### 3.3. Antimicrobial Results

Table 2 shows the inhibition zone amounts for the groups and the controls against *S. mutans*. The one-way ANOVA outcomes presented that addition of curcumin nanoparticles to self-cured resin enlarged the diameter of inhibition zones for the groups against *S. mutans* (*p* < 0.0001). Based on Tukey’s post-hoc test a significant difference between all groups and the 0% group in both bacteria was observed, which means that even 0.5 percent of curcumin nanoparticles shows the weighty antimicrobial act compared to the 0% group (*p* = 0.002).

Brandão et al. assessed the action against the *S. mutans* biofilm of model ZnO containing resin nanocomposites and described their physicochemical possessions. Based on their outcomes, as the concentration of ZnO nanoparticles increased, the antimicrobial action of the curing composite discs enhanced against *S. mutans* biofilms without any alteration on their physicochemical properties [30]. In another study, Morales et al. studied the antibacterial action of increasing concentrations of ZnO nanoparticles added to self-cured acrylic samples against on *Streptococcus mutans*. Their results also showed an increase in antibacterial action with the enhancing of the concentrations of ZnO nanoparticles [14]. The study by Esmaeilzadeh et al. showed that the adding of 2 and 4 wt% of ZnO, TiO_2_, and ZnO/TiO_2_ produced high antimicrobial effects comparing the control group (the resin without nanoparticles) [6].

It has been specified that it is better for the particle size of nanoparticles to be under 100 nm [16]. In the current study, the nanoparticles had an approximate size of 85 nm, which permits a greater interaction with the bacteria, which could affect the antibacterial result of the resins [14]. It has been verified by other investigators that there is a higher action for smaller particles than for larger ones owing to their high surface area, which leads to a large area for connections with bacterial cell walls [31]. However, in the current investigation, it cannot be confirmed that a quantity of them stay on the surface or in the center of the resin matrix. Then, the shape of nanoparticles may also have an impact on their bacteria interaction, such as rod-shaped particles that can enter the cell walls of bacteria more simply than spherical nanoparticles [31].

Based on the literature, curcumin hinders bacteria by injuring the bacterial membrane [15]. It has also been reported that curcumin may inhibit bacterial cell proliferation by perturbation of the FtsZ assembly. Some other investigations also established that curcumin can importantly deactivate bacteria by prompting the ROS production [17,18,32,33].

As well as its own functions, curcumin in nano-form can induce some other mechanisms of antimicrobial action. As mentioned earlier, one of the chief concerns for the clinical use of curcumin is its low solubility as well as itslow bioavailability. New nanoformulations are being created to improve curcumin’s bioavailability and reduce its probable toxicity in high doses [17,18]. Indeed, based on the particle properties (size, shape, surface charge) and type of bacteria (Gram-negative, Gram-positive, anaerobic, aerobic), nanoparticles induce their antibacterial action on bacteria by numerous mechanisms. The curcumin nanoparticles applied in this research had a mean particle size of 85 nm. The studies have exposed the chief part of physicochemical possessions and doses of nanoparticles in their antimicrobial activity [14,18]. There are some main antimicrobial mechanisms related to nanoparticles; the smaller nanoparticles can disorder bacteria cell membrane functions by binding to the surface of cell membranes with a high affinity compared to larger nanoparticles due to their larger surface area [16]. Additionally, the contact of bacteria’s membrane and nanoparticles occurs in the local pores in the bacteria cell membrane. Furthermore, the entrance of the nanoparticles into the bacteria also induce damage in DNA. Other studies have also reported that the nanoparticles can connect to the bacterial membrane and gradually enter the cytoplasm to disrupt bacterial main functions. Some types of nanoparticles may also fuse with the bacterial wall to release their content into the bacteria [14,18]. Hu et al. reported that curcumin decreases *Streptococcus mutans* biofilm formation through hindering sortase A action [34]. Additionally, Li et al. reported the antibiofilm effect of curcumin on the clinically isolated *S. mutans*. Their results showed that after 24 h curcumin treatment, the genes of gtfB, gtfC, gtfD, ftf, gbpB, fruA, and srtA were downregulated [35].

Pourhajibagher et al. assessed the antimicrobial actions of the curcumin-Nisin-poly (L-lactic acid) nanoparticle in orthodontic acrylic resin against *Streptococcus mutans* and *Candida albicans*. Based on their results, resin containing 5% of curcumin nanoparticles had a large inhibitory zone against the tested microorganisms. The authors believed that the decrease in the growth inhibition zones of the different concentrations of resin containing 5% of curcumin nanoparticles against tested microorganisms was related to the time and it was reduced significantly after 60 days [36].

In other study, Khamooshi et al. tested the antimicrobial action of crylic resin containing curcumin nanoparticles against *Streptococcus mutans, Streptococcus sanguinis, Lactobacillus acidophilus,* and *Candida albicans*. According to their results, none of the curcumin nanoparticle concentrations showed growth inhibition zones for any microorganisms. However, all the concentrations were effective against all four microorganisms in the biofilm inhibition test except 0.5% for *L. acidophilus* [37].

### 3.4. Cytotoxicity Assay

All groups were non-cytotoxic against dental pulp stem cells (Figure 5). According to ISO 10993-5, which sets the rules for the biocompatibility of medical devices, the acceptable level of cytotoxicity for medical devices is as follows:

If the number of living cells is ≥80%, the sample is free of cytotoxicity.

If the number of live cells is equal to 80%, the sample has moderate cytotoxicity.

If the number of living cells is ≥40%, the sample has high cytotoxicity [26].

### 3.5. Flexural Strength

Table 3 shows that as the weight percentage of curcumin nanoparticles increased, the flexural strength decreased (*p* < 0.0001). However, all strength values were higher than the standard value (50 MPa). No significant difference was detected between the control group and the group with a 0.5 percent weight of curcumin nanoparticles (*p* = 0.57).

Based on other investigators, the addition of curcumin nanoparticles to methyl methacrylate polymers increases their resistance against electrolytes penetration and creates impurities in the polymer, which can mainly be attributed to the presence of a hydroxyl functional group in curcumin structure. This functional group reinforces the hydrogen bonds between polymer monomers and leads to the reduction of the flexural strength [22].

Results of Shahabi et al. study indicated that adding a weight percentage of 1% chitosan nanoparticles reduces the flexural strength of self-cured acrylic resins [38]. Moslehi Fard et al. in their article on titanium nanoparticles observed that using 0.5%, 1%, and 2% weight percentages of this nanoparticle do not significantly affect the flexural strength of self-cured acrylic resins [39]. In another study, Sodagar et al. examined the impact of propolis nanoparticles (prpNPs) on antimicrobial action and shear bond strength (SBS) of orthodontic composite bonded to bovine enamel. The growth of *S. mutans* and *S. sanguinis* at all concentrations (except for 1%) was meaningfully lower than the control sample. The growth of *L. acidophilus* meaningfully decreased at 5% and 10% concentrations. The growth inhibition zone was bigger at 2%, 5%, and 10% concentrations for *S. mutans* and *S. sanguinis*. The lowest numbers of *S. mutans* and *S. sanguinis* colonies at all concentrations were detected on the 15th day. *L. acidophilus* colonies were reduced meaningfully at all concentrations (except for 1%) until the 30th day. They reported that in the weight percentages of 0.5%, 1%, and 2% they do not significantly affect the flexural strength of self-cured acrylic resin [23]. Esmaeilzadeh et al. examined the antibacterial and mechanical properties of a curing PMMA resin containing ZnO and TiO_2_ nanoparticles. Their obtained results showed that the addition of 2 and 4 wt% of ZnO, TiO_2_, and ZnO/TiO_2_ produced high antimicrobial effects. However, their results showed that the mean flexural strengths of acrylic specimens containing 2 wt% and 4 wt% of the used nanoparticles were meaningfully lower than the control group (resin without nanoparticles). Similar to our results, the samples indicated promising antibacterial possessions but decreased flexural strength [6].

In this study, the results also showed no significant difference between the control group and the group with a 0.5 percent weight of curcumin nanoparticles (*p* = 0.57). Our research has shown that the mixing method might affect the way nanoparticles dispersed in the matrix. Moreover, the lower weight percentages of nanoparticles may display better mixing with the resin matrix due to the low concentration and low possibility of agglomeration [40]. Moslehi Fard et al. considered the compressive, flexural, and impact strengths of heat-cured acrylic resins reinforced by TiO_2_ nanoparticles. In their study on TiO_2_ nanoparticles using electron microscope images indicated that the lower weight percentages of nanoparticles will show better sorting and homogenization, and fewer cracks will appear. The sample with 1 wt% TiO_2_ nanoparticles showed fewer micro-pores and micro-cracks in the SEM results. According to authors, a non-significant growth was also obtained in the impact strength with TiO_2_ nanoparticles at 1 wt%. An additional increase in TiO_2_ nanoparticles reduced both the impact and flexural strengths. The authors concluded that the compressive strength of the acrylic resin was not influenced by the incorporation of TiO_2_ nanoparticles [39].

Pourhajibagher et al. also evaluated the mechanical properties of acrylic resins containing curcumin-Nisin-poly(L-lactic acid) nanoparticle. Acrylic resin with 5% nanoparticles showed clinically accepted flexural strength value (14.89 ± 3.26 MPa, *p* < 0.05) [36].

A study by Comeau et al. evaluated curcumin-containing acrylic resins (0.05 and 0.10 wt%). The outcomes confirmed that 0.10 wt% curcumin had minimal impact on either flexural strength or shear bond strength, but obviously increased water sorption and solubility. It also showed good antimicrobial effects against *S. mutans* [41].

## 4. Limitations of the Study

This study did not investigate the polymerization degree of modified samples, which affects the physical possessions of acrylic resin [42]. Photographic analysis of cracked surfaces was not also carried out. Further studies should be conducted to determine the effect of the curcumin nanoparticles on the mechanical and physical properties of self-cured acrylics. The color change and cytotoxicity of curcumin nanoparticles should also be investigated in different in vitro and in vivo levels.

## 5. Conclusions

In this study, we tested the release study, the antimicrobial effects, and the flexural strength changes of self-cured acrylic resins after the addition of curcumin nanoparticles. The release patterns presented a two-stage release outline for the nanocurcumin-containing resins. The obtained antimicrobial outcomes exhibited that all resin samples (with different percentage of nanocurcumin) had antimicrobial possessions against *Streptococcus mutans* (*S. mutans*). Furthermore, as the concentration of the curcumin nanoparticles increased, the flexural strength reduced. Nevertheless, all strength values were higher than the standard value (50 MPa). Therefore, considering the potent antimicrobial activity of curcumin nanoparticles and its sustained release in this form, it seems that formulating self-cured acrylic resins containing curcumin nanoparticles can be helpful for antimicrobial aims without having destructive effects on the flexural strength of them to use in orthodontic removable applications. More investigations are needed to show the real and usable clinical effect of the curcumin nanoparticles on the other properties of self-cured acrylics. The color alteration and the cytotoxicity tests of curcumin nanoparticles should also be examined.

## 6. The Future Perspectives

Nanomaterials have been found to be more effective than conventional-size materials and to present greater abilities in terms of surface modifications. Antimicrobial nanoparticles are useful in all fields of dental procedures, mostly in endodontics and orthodontics. Due to their versatility, they are a beneficial tool in clinical dentistry for a range of applications. Some investigations on nanomaterials have found considerable consequences. There are some types of nanomaterials that have led to industrial manufacture. However, studies on the detail of the action mechanisms and establishment procedure of the microstructure of nanomaterials still remain poor, and many parts still essential to be discovered. On the other hand, the broad application forecasts and huge potential value in future dentistry will be determined due to the unique properties of nanomaterials. Thus, it is necessary to keep developing new nanomaterials and expand our understanding of their mechanism, strengthening procedure, and alteration procedures to improve their possessions. Developing novel formulations of plant-based antimicrobial nanomaterials that together with their antimicrobial action can keep or increase the mechanical properties of acrylic resins can be the other future aspects in this regard. Furthermore, based on previous studies, seeing the association between orthodontic cure and gingival healthiness, team work among orthodontists, periodontists, and patients is essential.

## Figures and Tables

**Figure 1 bioengineering-10-00559-f001:**
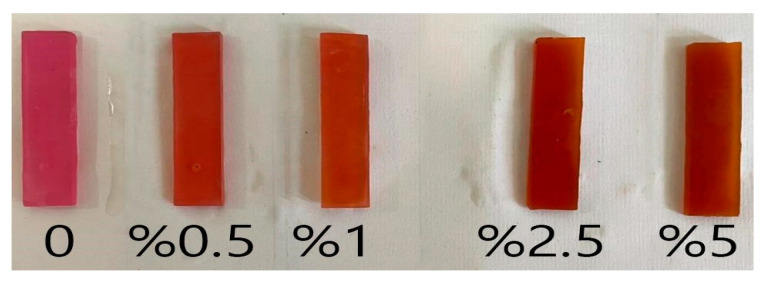
Modified acrylic resin samples with varied weight percentage of curcumin nanoparticles (0 for control, 0.5, 1, 2.5, and 5%).

**Figure 2 bioengineering-10-00559-f002:**
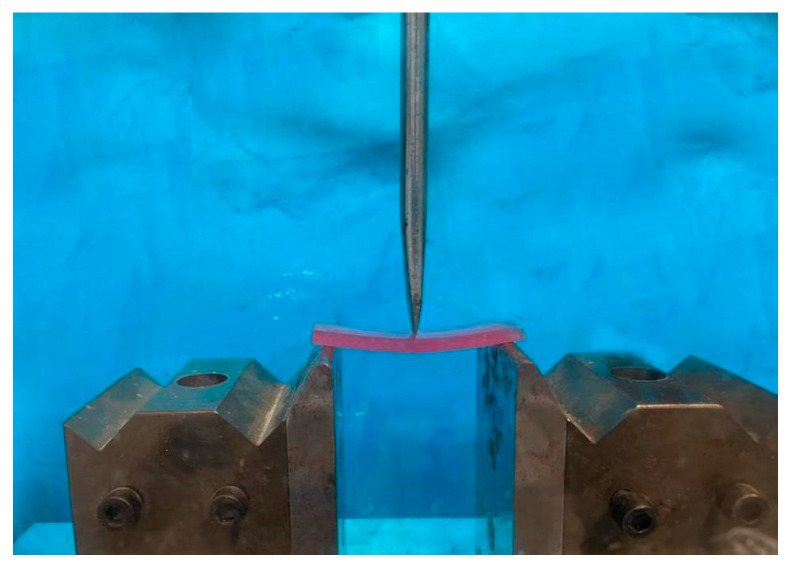
Samples were performed the 3-point flexural strength test.

**Figure 3 bioengineering-10-00559-f003:**
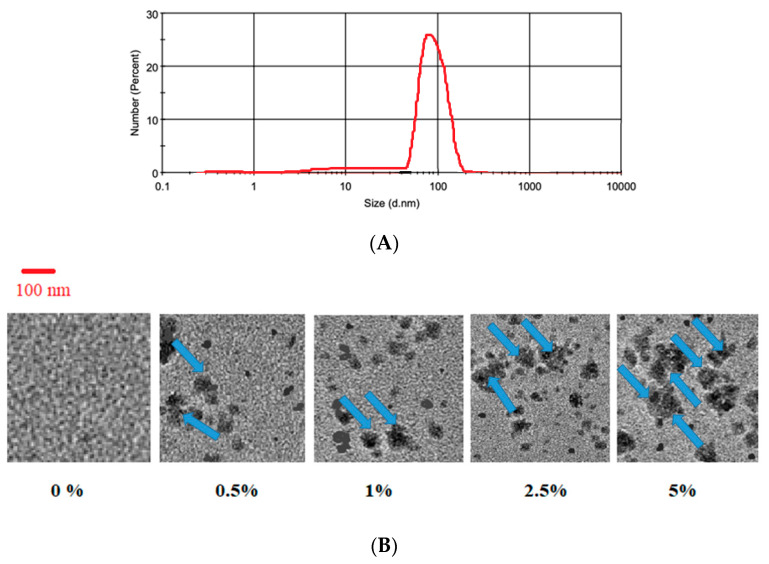
(**A**) The size distribution for curcumin nanoparticle. (**B**) The TEM images of self-cured acrylic resins modified with varied nanocurcumin concentrations. Compared with the TEM image of group 1 (0 percent nanoparticles), the nanoparticles’ dispersion is completely clear in the other groups. All the aggregated dispersion in the images for 0.5, 1, 2.5, and 5 percent groups are curcumin nanoparticles (Blue arrows).

**Figure 4 bioengineering-10-00559-f004:**
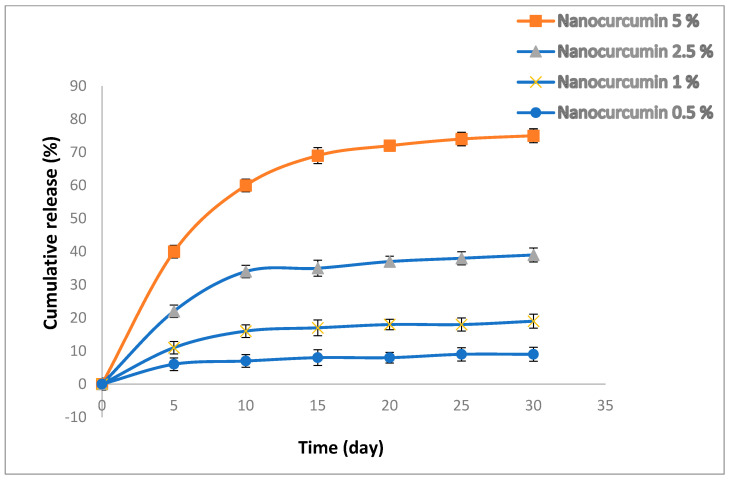
The release profile of the nanocurcumin from resins.

**Figure 5 bioengineering-10-00559-f005:**
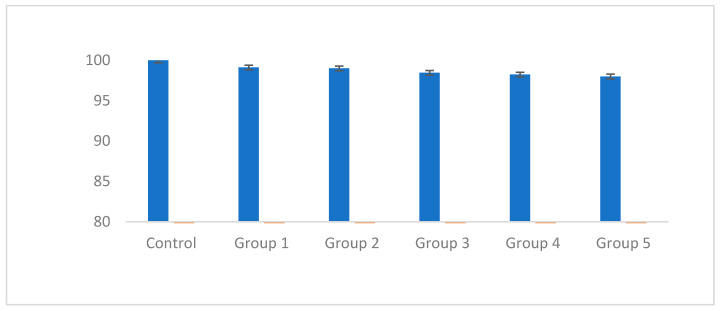
The cytotoxicity results of the groups against dental pulp stem cell.

**Table 1 bioengineering-10-00559-t001:** The formulations for the samples.

Groups	Formulations
Group 1, 0 percent curcumin nanoparticles (n = 12)	0 g curcumin nanoparticles, 5 g acrylic resin and 3.5 g monomer
Group 2, 0.5 percent curcumin nanoparticles (n = 12)	0.0425 g curcumin nanoparticles, 5 g acrylic resin and 3.5 g monomer
Group 3, 1 percent curcumin nanoparticles (n = 12)	0.085 g curcumin nanoparticles, 5 g acrylic resin and 3.5 g monomer
Group 4, 2.5 percent curcumin nanoparticles (n = 12)	0.2125 g curcumin nanoparticles, 5 g acrylic resin and 3.5 g monomer
Group 5, 5 percent curcumin nanoparticles (n = 12)	0.425 g curcumin nanoparticles, 5 g acrylic resin and 3.5 g monomer

**Table 2 bioengineering-10-00559-t002:** The inhibition zone sizes for the groups against *S. mutans*.

Samples	Inhibition Zone (mm)
	*S. mutans*
0% curcumin NPs (negative control)	0
0.5% curcumin NPs	3.80 ± 0.08
1% curcumin NPs	9.33 ± 0.62
2.5% curcumin NPs	12.36 ± 0.63
5% curcumin NPs	16.00 ± 0.81
Vancomycin (positive control)	17.2 ± 1.50

**Table 3 bioengineering-10-00559-t003:** Mean flexural strength of self-cured acrylic resin modified with different weight percentages of nanocuricumin.

Samples	Mean Flexural Strength
0% curcumin NPs (negative control)	60.23 ± 1.35
0.5% curcumin NPs	59.80 ± 1.05
1% curcumin NPs	54.58 ± 1.00
2.5% curcumin NPs	52.36 ± 1.07
5% curcumin NPs	50.03 ± 1.81

## Data Availability

The data for the current study can be shared at this time.

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
