# Peer review of "Nanocurcumin Release from Self-Cured Acrylic Resins; Effects on Antimicrobial Action and Flexural Strength"

_bioengineering, 2023, doi:10.3390/bioengineering10050559_

Round 1
Reviewer 1 Report
The manuscript is quite interesting. However, there are some points which should be completed and clarified, as listed below:
1. The authors should separate the materials section to explain all materials used and their orgin.
2. A formulation table is required to improve the readibility of this manuscript. The authors should also explain the scientific reason on choosing the different concentration of nanoparticles as the variable in this research.
3. The authors did not mention which samples were used in the microscopy study. Since the product resulted was in a dry form, the authos should also perform SEM study.
4. The authors should also provide nanoparticle size distribution measured by Particle Size Analyzer.
5. Release pattern must be replaced with release study.
6. The release study was conducted until 30 days, however there is no mehod validation for the UV Spectrophotometer methods, also a stability study is also important to perform.
7. References are needed in some methods.
8. Figure 3 should be completed with anotation showing the nanoparticles.
9. Some typos in Figure 4 legend: nnaocurcumin.
10. The authors should also perform some important studies mentioned in the Limitation of Study for improving the quality of this manuscript, such as: the polymerization degree of modified samples, morphological study of samples fabricated and cracked surfaces, the curcumin nanoparticles on the mechanical and physical properties of self-cured acrylics, stability study of the prepared samples, cytotoxicity of curcumin nanoparticles (at least in vitro study).
Author Response
The manuscript is quite interesting. However, there are some points which should be completed and clarified, as listed below:
1. The authors should separate the materials section to explain all materials used and their origin.
Thanks for your valuable comments. It has been done.
A formulation table is required to improve the readibility of this manuscript. The authors should also explain the scientific reason on choosing the different concentration of nanoparticles as the variable in this research.
Thanks for your valuable comments. The concentrations were chosen based on pilot study according to free curcumin nanoparticle's inhibitory zone diameter. A table has been added.
The authors did not mention which samples were used in the microscopy study. Since the product resulted was in a dry form, the authors should also perform SEM study.
Thanks for your valuable comments. For TEM analysis, 3 samples were chosen randomly from each group. Then, the best Figure from quality view was chosen to inserted in the manuscript. We had SEM images. However, SEM images did not give information about dispersion of curcumin containing acrylics. They should be in powder form.
The authors should also provide nanoparticle size distribution measured by Particle Size Analyzer.
It has been done.
Release pattern must be replaced with release study.
It has been done.
The release study was conducted until 30 days, however there is no method validation for the UV Spectrophotometer methods, also a stability study is also important to perform.
It has been added.
7. References are needed in some methods.
It has been added.
Figure 3 should be completed with anotation showing the nanoparticles.
Thanks for your valuable suggestion. Comparing with TEM image of group 1 (0 percent nanoparticles), the nanoparticles dispersion is completely clear in the other groups. All the aggregated dispersion in the images are nanoparticles. We added this into the figure legend.
- Some typos in Figure 4 legend: nnaocurcumin.
It has been corrected.
The authors should also perform some important studies mentioned in the Limitation of Study for improving the quality of this manuscript, such as: the polymerization degree of modified samples, morphological study of samples fabricated and cracked surfaces, the curcumin nanoparticles on the mechanical and physical properties of self-cured acrylics, stability study of the prepared samples, cytotoxicity of curcumin nanoparticles (at least in vitro study).
We added some new data.
Reviewer 2 Report
The work is good. But needs minor revision in t1. Figure 4. The release pattern of the nanocurcumin from resins. he following points. needs corrections for reducing the size of legends so that the error bars are visible clearly in all the groups. In nanocurcumin 5 %curve the error bars are not completely visible and in others it is not clear completely. Either reduce the size or make some legend without filling of color
2. Curcumin as an antimicrobial is well known, the self-curing acrylic resins, and their use with curcumin is also not new. Moreover, I could not find citations and discussions of previous studies on the same work. Some of the studies which have done similar work have not been discussed in the discussion.
1. Khamooshi P, Pourhajibagher M, Sodagar A, Bahador A, Ahmadi B, Arab S. Antibacterial properties of an acrylic resin containing curcumin nanoparticles: An in vitro study. J Dent Res Dent Clin Dent Prospects. 2022 Summer;16(3):190-195. doi: 10.34172/joddd.2022.032.
2. Pourhajibagher, M., Noroozian, M., Ahmad Akhoundi, M.S. et al. Antimicrobial effects and mechanical properties of poly(methyl methacrylate) as an orthodontic acrylic resin containing Curcumin-Nisin-poly(L-lactic acid) nanoparticle: an in vitro study. BMC Oral Health 22, 158 (2022). https://doi.org/10.1186/s12903-022-02197-z
3. Comeau, P., Panariello, B., Duarte, S. et al. Impact of curcumin loading on the physicochemical, mechanical and antimicrobial properties of a methacrylate-based experimental dental resin. Sci Rep 12, 18691 (2022). https://doi.org/10.1038/s41598-022-21363-5
Please include these studies in the discussion and discuss your results in reference to these studies and other similar studies.
Author Response
The work is good. But needs minor revision in t1. Figure 4. The release pattern of the nanocurcumin from resins. he following points. needs corrections for reducing the size of legends so that the error bars are visible clearly in all the groups. In nanocurcumin 5 %curve the error bars are not completely visible and in others it is not clear completely. Either reduce the size or make some legend without filling of color
Thanks for your valuable comments. We improved the Figure.
- Curcumin as an antimicrobial is well known, the self-curing acrylic resins, and their use with curcumin is also not new. Moreover, I could not find citations and discussions of previous studies on the same work. Some of the studies which have done similar work have not been discussed in the discussion.
- Khamooshi P, Pourhajibagher M, Sodagar A, Bahador A, Ahmadi B, Arab S. Antibacterial properties of an acrylic resin containing curcumin nanoparticles: An in vitro study. J Dent Res Dent Clin Dent Prospects. 2022 Summer;16(3):190-195. doi: 10.34172/joddd.2022.032.
- Pourhajibagher, M., Noroozian, M., Ahmad Akhoundi, M.S. et al.Antimicrobial effects and mechanical properties of poly(methyl methacrylate) as an orthodontic acrylic resin containing Curcumin-Nisin-poly(L-lactic acid) nanoparticle: an in vitro study. BMC Oral Health 22, 158 (2022). https://doi.org/10.1186/s12903-022-02197-z
- Comeau, P., Panariello, B., Duarte, S. et al.Impact of curcumin loading on the physicochemical, mechanical and antimicrobial properties of a methacrylate-based experimental dental resin. Sci Rep 12, 18691 (2022). https://doi.org/10.1038/s41598-022-21363-5
Please include these studies in the discussion and discuss your results in reference to these studies and other similar studies.
It has been done. Thanks.
Reviewer 3 Report
This study evaluates the release pattern, the antimicrobial action and the flexural strength of self-cured acrylic resins after adding different weight percentages of curcumin nanoparticles (nanocurcumin). The topic of the manuscript is very important, because the antibacterial nano particles play role in the long-term durability of restorations and protection of oral hygene. The manuscript is well designed and written. It gives informative and clear information. The results are confirmative of the topic.
Author Response
Thanks for your valuable comments.
Round 2
Reviewer 1 Report
The authors have addressed reviewer comments very well. Moreover, the quality of the manuscript has been improved by adding some important data, as per requested. However, some points should be revised, as listed below:
1. the "release pattern" in the Figure 4 caption has not been replaced with "release profile". Also, the font size of the Figure 4 is too small.
2. there is no annotation/arrow in Figure 3 to point the nanoparticles.
3. The LOQ and LOD should also be mentioned in the method validation result.
I recommend this manuscript for publication in Bioengineering after the minor correction mentioned above.
Author Response
Thanks for your valuable suggestions.
- the "release pattern" in the Figure 4 caption has not been replaced with "release profile". Also, the font size of the Figure 4 is too small.
It has been done.
- there is no annotation/arrow in Figure 3 to point the nanoparticles.
It has been added.
- The LOQ and LOD should also be mentioned in the method validation result.
It has been added.
I recommend this manuscript for publication in Bioengineering after the minor correction mentioned above.
Thanks a lot